🔓 PLOS | ONE

# Experiences with peer support for breastfeeding in Beirut, Lebanon: A qualitative study

Tamar Kabakian-Khasholian[1], Hana Nimer[1], Soumaya Ayash[2], Fatima Nasser[2], Mona Nabulsi🅾[2]*

**1** Department of Health Promotion and Community Health, Faculty of Health Sciences, American University of Beirut, Beirut, Lebanon, **2** Department of Pediatrics and Adolescent Medicine, American University of Beirut, Beirut, Lebanon

* mn04@aub.edu.lb

## Abstract

### Background

Despite the beneficial effects of peer support on breastfeeding, research on the process of peer support is scarce. In Lebanon, exclusive breastfeeding is only 15% in infants below six months. A multidisciplinary team launched a multi-component breastfeeding support intervention, with peer support, and professional lactation support provided by International Board Certified Lactation Consultants (IBCLs) to target this decline.

### Aim

To describe the experiences of breastfeeding mothers and peer support providers with the process of breastfeeding support, and the influence of the intervention on their social support system.

### Methods

Using a qualitative methodology, a purposive sample of breastfeeding and support mothers was accessed from among those who completed their six months interview in the trial taking place in two hospitals in Beirut, Lebanon. Data were collected from 43 participants using in-depth interviews and following the data saturation principle. All interviews were audio recorded and transcribed verbatim. Thematic analysis was conducted, guided by the principles of grounded theory.

### Results

Breastfeeding mothers were satisfied with their breastfeeding experience, and extremely appreciative of the support provided by their peers and the IBCLCs. They experienced these forms of support differently. Peer support was perceived to be important in encouraging breastfeeding continuation, whereas IBCLC support was influential in problem solving.

**Data Availability Statement:** All relevant data are within the manuscript and its Supporting Information files.

**Funding:** This study was funded by a grant awarded to Dr. Mona Nabulsi from the Medical Practice Plan of the Faculty of Medicine at the American University of Beirut (MPP 11485.720.999). The funder had no role in the study design, data collection and analysis, decision to publish, or preparation of the manuscript.

**Competing interests:** The authors have declared that no competing interests exist.

## Conclusion

These findings can improve our understanding of the peer and professional lactation support process within the social context, and the realities of breastfeeding mothers, and help facilitate the scaling up of interventions in similar contexts.

## Introduction

Studies and reviews of breastfeeding interventions suggest that provision of support for breastfeeding is effective in improving both short-, and long-term breastfeeding rates, especially when this support is provided continuously from pregnancy throughout the postnatal period, and is provided by peer supporters, by lactation professionals, or a combination of both [1]. Similar evidence from low or middle income countries reveals that peer support improved the duration of exclusive breastfeeding [2]. Moreover, the cumulative effect of peer and professional lactation support improves exclusive breastfeeding practices, especially when interventions include face-to-face components [1].

There is a wealth of knowledge about the effectiveness of breastfeeding support, and specifically of peer support for breastfeeding [1, 3]. Breastfeeding peer support is provided to mothers by women volunteers in their communities. The peer supporters usually have previous experience in breastfeeding, with or without some form of informal training. Despite the recent evidence on the content of, and form in which peer support should be provided to improve its effectiveness [2], there are still gaps in our understanding of how breastfeeding mothers use peer support, and how their experience with receiving peer support benefits breastfeeding practices. Moreover, the understanding of how mothers experience the form of support provided to them during a clinical trial may facilitate the implementation of the intervention at a larger scale, and will help in the interpretation of participants' responses to the different aspects of the intervention package in the trial [4].

Breastfeeding mothers need different types of support that are summarized in the following five forms: informative, emotional, face-to-face, instructional, and self-support [5, 6]. Mothers receive mixed messages about breastfeeding from their social circles, including families and health care providers. Therefore, building self-efficacy through provision of support becomes important [7]. Considering that breastfeeding is a socially constructed practice, it is expected that breastfeeding peer support will influence, and further develop the social network of breastfeeding mothers into new dimensions in terms of communication processes and social support. The social network and social support approach we used to understand health behaviors in this study describes several ways through which social ties and relationships can influence health, or health behaviors. The relationships in a network that support recommended practices enhance health and well-being by overcoming the stressors, and enhancing the individual's ability to access new contacts and information; and to identify and manage problems. The different types of support provided by the the IBCLCs, and by the peer supporters can fulfil mothers' needs for information, emotional support, and face-to-face encounters to listen to her concerns, as well as instructional support in the form of practical advice and help. These forms of support can also enhance self-efficacy of the mother, and strengthen her ability to self-support [5].

Peer support in breastfeeding builds social capital by connecting women who have similarities in closed networks, and in some instances, by bridging women from different economic, ethnic and professional backgrounds [8]. However, some barriers can impede mothers from

benefiting from peer support, such as being unaware of the available breastfeeding peer support programs, not knowing what a supporter does, being uncomfortable with meeting with an unfamiliar person, seeking other forms of support like online social networking, and not asking for help [9].

Peer support is also beneficial to the supporters themselves who, by undertaking a specific training, develop supporting skills, and may use the training as proof of achievement, and a platform for employment and further training in the health and social sectors [10]. Moreover, peer supporters may engage with community, health, political, statutory, and commercial sectors in order to spread awareness of peer support and breastfeeding, which will help them reach mothers, and promote a positive attitude towards breastfeeding in the community [8].

In Lebanon, the rate of exclusive breastfeeding is only 15% in infants below six months of age [11]. Despite universal coverage and use of prenatal care and hospital births [12], there are no structured activities, or interventions that are embedded within the health care system that specifically aim at providing information and support for breastfeeding mothers. A multidisciplinary group of researchers at the American University of Beirut launched a multi-component breastfeeding support intervention consisting of breastfeeding education, peer support, and professional lactation support delivered by IBCLCs to target this gap. This intervention aimed at improving mothers' breastfeeding knowledge, skills, and social support to increase exclusive breastfeeding duration [13]. The trial is registered in Current Controlled Trials ISRCTN17875591. www.isrctn.com. The multi-component intervention was effective in improving the rate of six-month exclusive breastfeeding in the intervention group by two folds, as compared to standard care [14].

This paper reports the findings from a nested study within the breastfeeding support trial [13, 14]. It aims to describe the experiences of providing, and receiving peer support from the perspective of breastfeeding mothers and peer supporters. It provides an in-depth description of the process of peer support, and the mechanisms through which this type of support is perceived to influence the social context, and the realities of breastfeeding mothers and their peer supporters. The results of this study can inform the tailoring of peer support training programs in similar contexts, and the scaling up of similar interventions.

## Materials and methods

### Design

This study is nested within a randomized controlled clinical trial that was conducted in two tertiary care hospitals in Beirut [13, 14]. It is a cross-sectional, prospective, two group qualitative design. Qualitative research methodologies enable the researcher to capture the way individuals react, understand, and think about the questions through the dynamics of the situation, and the perspective of the individual who is living it [15]. This study conforms to the Consolidated Criteria for Reporting Qualitative Studies (COREQ) [16].

The trial enrolled healthy pregnant women in their first or second trimesters, who expressed intention to breastfeed after giving birth. Participants in the experimental group received a multi-component intervention consisting of: (a) prenatal breastfeeding education delivered by IBCLCs in one of the two trial centers as prenatal classes to raise awareness, and improve knowledge, (b) postpartum lactation support provided by IBCLCs to improve skills, and self-efficacy through home visits, (c) postpartum peer (lay) telephone support to build social support, and enhance social capital. The primary outcome of the trial, exclusive breastfeeding, was assessed at the six month interview with the mothers. More details on the trial's multi-component intervention and the trial's findings can be found elsewhere [13, 14].

Peer supporters were recruited via three methods: (1) each participant in the trial's intervention group was asked to identify one to three women in her community or family who could serve as peer supporters; (2) flyers inviting women to become peer supporters were posted on bulletin boards of pediatric, and obstetric clinics in the participating hospitals; (3) snowball effect: enrolled peer supporters were asked to identify one to three women in their communities/families who could serve as peer supporters. Those enrolled via methods (2) or (3) above were matched with intervention trial participants who could not identify potential peer supporters from their community, or with participants who nominated peer supporters that were later judged by the research team not to meet the inclusion criteria of a peer supporter. The matching process was based on age, availability, and geographical proximity to the trial participant. In total, 39 peer supporters were recruited into the trial. The number of breastfeeding mothers assigned to the same peer supporter ranged between 1 and 9, depending on availability of peer supporter. Some of the breastfeeding mothers were known to peer supporters through their social networks, while others were not.

In order to be eligible for enrolment as a peer supporter, the volunteering woman had to have successfully breastfed at least one child for two months, have a positive attitude towards breastfeeding, be able to attend two half day training sessions to learn how to support new mothers, and when to refer to professional resources; and be able to read and write in Arabic, at middle school level or higher. Training workshops for peer supporters were conducted by one of three pediatricians who were part of the trial's research team. Each workshop lasted for four hours and included a maximum of ten peer supporters. The first workshop covered a brief overview of the peer support program: training on the *listen-observe-validate empower/educate (LOVE)* method of support [17]; discussion of *breastfeeding basics*, including advantages of human milk and risks of artificial milk, common culture-specific misconceptions, and demonstration of the proper breastfeeding technique using visual aid material. Moreover, breastfeeding trouble shooting, and criteria for referral to appropriate medical services were emphasized. The second workshop was largely practical. Peer supporters engaged in interactive discussions and role playing of "*what if*" scenarios as necessary, including failure to breastfeed scenarios, being empathetic, and showing a positive attitude. Subsequently, peer supporters were provided with a manual containing a list of local breastfeeding resources, as well as logbooks in which they had to record the details of their communications with the breastfeeding mothers, and the kind of support provided. Peer breastfeeding support occurred in an informal manner based on a minimum number of ten scheduled calls, or hospital/home visits, starting with the antenatal class, then at the sixth and ninth months of gestation, the expected week of delivery, the first day postpartum, 48 hours from hospital discharge, one, two, and four weeks postpartum, and monthly thereafter until six months postpartum. This schedule could be modified based on the needs of the breastfeeding mother. For logistical reasons, peer support ended up being delivered post-natal and as telephone calls. Peer support continued until the infant was six months of age, or until the breastfeeding mother stopped breastfeeding, or withdrew from the clinical trial, whichever came first [13, 14].

Professional lactation support was delivered by IBCLCs who visited the participants according to a pre-specified schedule that started on the first postpartum day in the hospital, and continued as daily visits of 15–30 minutes until the woman is discharged from the hospital. IBCLCs conducted home visits on days 1, 3, 7 and 15 and then monthly until six months, until breastfeeding discontinuation, or maternal request to stop, whichever came first. IBCLC support was provided mainly as face-to-face, but could also happen via telephone if so requested by the participant. Additional visits were permitted if requested by the mother, or judged to be necessary by the IBCLC [13, 14].

The trial enrolled 174 participants in the experimental group, and 188 in the control group. The total number of peer supporters enrolled in the trial was 39. Participants in the control group received only standard prenatal and postnatal care as dictated by their obstetricians and pediatricians at both sites. Optional prenatal classes about labor, delivery, and breastfeeding were available at one trial site, but not at the other one. IBCLCs are unavailable at both centres.

## Setting

In Lebanon, both prenatal and postnatal care does not follow standard practices, and attempts to include breastfeeding practices in the care system are completely dependent on individual care providers in a highly privatized health system. Although prenatal care is widely utilized with 96% reporting at least one prenatal visit, the six week postnatal checkup, the only form of providing postnatal care, is used by only around half of the birthing population [12]. This fragmented health care system may potentially contribute to the low exclusive breastfeeding rate of 15% in infants below six months in this country [12]. Information on infant feeding and nutrition is usually provided by pediatricians and hospital nurses. Hospitals and maternities in Lebanon do not comply with the World Health Organization's (WHO) ten steps of Baby-Friendly Hospital Initiative, and health professionals who care for breastfeeding mothers lack the WHO, and UNICEF recommended training in the prevention and treatment of breast-feeding problems [18, 19]. Separation of mother and infants is practiced at hospitals throughout their stay, and artificial milk is available, and is used with infants in these hospitals. The few IBCLCs in Lebanon have private practices, and they are not employed by hospitals.

The Global Strategy for Infant and Young Child Feeding has been endorsed in Lebanon in 2002. This was followed in 2008 by a law that was enacted to regulate marketing of breast milk substitutes. The country however faces challenges in enforcement of the law and implementation of strategies [20].

The two hospitals where the trial was conducted are both private. One is a large teaching private hospital located in the capital Beirut. It is used by women from different regions of the country as a reputable referral center. The other is a small tertiary hospital located in the southern suburb of Beirut. It caters mainly to the communities residing in its neighborhood characterized by middle to low socioeconomic status.

## Sample

Participants were a purposive sample of breastfeeding mothers, and peer supporters who completed six months in the trial. The main inclusion criteria were: the participant had approved to be contacted for future studies in the trial's original consent form, and expressed interest in a follow up interview. The principle of saturation of new emerging themes was used to indicate when to stop data collection. This was achieved with 22 breastfeeding mothers and 21 peer supporters. The purpose of the selection process of the mothers was to ensure variability in terms of parity, educational level, employment status, hospital site, and success in exclusive breastfeeding for six months. A similar approach was adopted for selecting peer supporters to ensure variation in the neighborhoods, employment status, and educational level. IBCLCs were not interviewed in this study.

## Data collection

The data collection period extended from May 2015 to December 2016. A trained research assistant contacted breastfeeding mothers and peer supporters, and arranged for appointments to conduct in-depth interviews at their homes. The research assistant was trained in obtaining informed written consent from all participants for interviewing, and recording of the

interview. The interview guides had different questions for the breastfeeding mothers, and for the peer supporters (S1 Appendix). The confidentiality of participants' responses, as well as their anonymity was preserved by using serial numbers and pseudonyms on transcripts of interviews, which will also help identify different respondents throughout the analysis, and reporting of findings. The audio recordings were used only by the study team, and were deleted after completion of the transcription process. The interviewer was a female who had experienced breastfeeding, and holds a degree in psychology. Her background and approach created a comfortable environment for the participants to discuss their breastfeeding journeys, as well as their struggles. Also, stating explicitly that she is a researcher in the team who is not involved in the trial was beneficial in encouraging participants to talk about the different aspects of their experiences with the trial. Credibility was enhanced through debriefing the primary investigator on the interviews, and discussing of emerging issues, and informing next set of interviews.

### Data analysis

The interviews were audio recorded, and transcribed verbatim by the same researcher. The grounded theory was used, and thus concurrent data collection and analysis was conducted. Thematic analysis was conducted manually, using the transcripts separately for breastfeeding mothers and peer supporters. Two authors (T.K. and H.N.) read and re-read the transcripts to identify, and refine emerging codes, and check for meanings. These codes were merged into subthemes and themes as common concepts emerged from the data. Some subthemes were also identified following the questions raised by the interviewer. Matrices were constructed to organize themes, and to compare the occurrence of themes across all respondents. Further analysis was conducted across different statements of respondents related to the same theme, as well as across different themes to identify possible connections [21]. The credibility of the interpretations was enhanced by sharing and discussing emerging themes with members of the study team.

## Results

### Characteristics of breastfeeding mothers and peer supporters

The age of breastfeeding mothers interviewed in this study ranged between 23 and 40 years, whereas that of the peer supporters ranged from 26 to 50 years. Table 1 details the characteristics of both breastfeeding mothers and their peer supporters.

### Themes

During the interviews, breastfeeding mothers drew comparisons between the different types of support they received from the peer supporters and the IBCLCs, describing the types and forms of support they received. Therefore, sub-themes were grouped under the major theme of the need for different types of support. The second theme emerged as a result of interview questions about the influence of receiving and providing breastfeeding support on participants' social networks. These were separately available for the two groups of participants. The third theme reflects participants' narratives on the influence of their involvement in the trial in becoming breastfeeding advocates. This emerged from the participants' enthusiasm in putting what they learned and experienced into use for others in their lives. This was also seen in both groups of participants (Table 2).

**The need for different types of support.** Breastfeeding mothers appreciated the support provided to them from the peer supporters, and from the IBCLCs. They attributed their

**Table 1. Characteristics of breastfeeding mothers and peer supporters (N = 43).**

| Characteristics | | Breastfeeding mothers (n = 22) n (%) | Peer supporters (n = 21) n (%) |
|---|---|---|---|
| **Hospital** | A | 14 (63.6) | 13 (61.9) |
| | B | 8 (36.4) | 8 (38.1) |
| **Number of children** | 1 | 15 (68.2) | 3 (14.3) |
| | 2–3 | 7 (31.8) | 12 (57.1) |
| | 4+ | 0 (0.0) | 6 (28.6) |
| **In paid work** | Yes | 11 (50.0) | 7 (33.3) |
| | No | 11 (50.0) | 14 (66.7) |
| **Education** | Intermediate or secondary | 8 (36.4) | 11 (52.4) |
| | University | 14 (63.6) | 10 (47.6) |
| **Breastfeeding duration** | < 6 months | 9 (40.9) | 0 (0.0) |
| | ≥ 6 months | 13 (59.1) | 21 (100.0) |

success in breastfeeding to their participation in the trial that provided them with different forms of support:

> *Without this study I could have stopped breastfeeding from the first month . . . Without the help of the peer support and consultant. If there is no one supporting you, you can easily give up, you think it is a big issue and you are not capable of handling it, but with this study you feel like you can breastfeed.*

> *(29 years old, one child)*

In general, mothers appreciated the support provided by the IBCLCs much more than the peer support. They described the IBCLCs as "credible" sources of information, praised their competence, and considered them as the focal person that they referred to throughout their breastfeeding journey. The IBCLCs' home visits were extremely valued by the breastfeeding mothers as they provided an opportunity for hands-on advice on problem solving that was very much needed, especially during the first weeks following birth:

**Table 2. Themes, theme definitions, and codes related to breastfeeding support.**

| Theme | Theme Definition | Codes |
|---|---|---|
| Experiences with types of support | Participants expressed their use of different forms of support they have received from peer supporters and from IBCLCs | **Type of support**—emotional support, help with breastfeeding, information, problem solving, encouragement to continue. **Form of support**–telephone calls, home visits, text messages. **Comparison**–peer supporter less needed than IBCLC, IBCLC more competent, preference for home visits, telephone calls when needed. |
| Social networks | For breastfeeding women: Participants did not enhance their social groups by including the peer supporters. | **Not face-to-face**—did not meet peer supporter, only telephone calls, meeting other breastfeeding women, peer supporters not present at prenatal sessions. |
| Social networks | For peer supporter: Peer supporters used the gained knowledge with breastfeeding and pregnant women in their social networks. | **Using knowledge outside of study**–reaching out to pregnant women, advising breastfeeding women, social gatherings, solicited by friends/family. |
| Breastfeeding advocates | For breastfeeding mothers: Breastfeeding mothers reached out to other mothers in their social circle. | **Breastfeeding advice to others**–talking about benefits, persuading to breastfeed, becoming a peer supporter, giving encouragement, helping others, want to talk about breastfeeding. |
| | For peer supporters: Peer supporters endorsed the role of breastfeeding advocates in their communities. | **Endorsing new role**–want to have more skills, want to continue, responsibility to inform others, benefiting from trial, recognition by others. |

*I had a stronger relationship with the consultant because her info was more scientific and she is an expert. . ..I saw her every month.*

*(30 years old, two children)*

*You know it is different when she is here and can show you positions to breastfeed. Then you understand what you are doing wrong. We need that. This is more important.*

*(29 years old, two children)*

The main contribution of peer supporters as perceived by the breastfeeding mothers was the provision of moral support. The mothers needed encouragement to continue breastfeeding when faced with difficulties. One participant expressed the positive role of her peer supporter when she was considering quitting breastfeeding as "*. . .the push you need to continue*". Peer supporters shared with the mothers they supported their own experience, and the experiences of other breastfeeding mothers, something that enticed these participants to continue breast-feeding, as they realized that these problems are expected, and that all breastfeeding mothers go through them at some point in time:

*To know that other people have gone through the same problems is a relieving thought by itself.*

*(32 years old, two children)*

*I sometimes feel like I will stop, and then as if she knows, she gives me a call. We talk, and then I feel much better and decide to continue a little bit more. Every time it's similar, she will call me, and I decide to continue.*

*(30 years old, three children)*

**Social networks.**   Breastfeeding mothers did not perceive any benefits of the peer support on their social network. The fact that they had not met their peer supporters in person was usually contrasted to the home visits paid by the IBCLCs. They viewed the lack of face-to-face contact with peer supporters as a major barrier to their inclusion within their social circle:

*I remember once or twice when she called me. She calls you but that is not enough to benefit from her. It is not like the one who visits you for half an hour and gives you instructions on positions and the infant care.*

*(26 years old, one child)*

Participants suggested having peer supporters pay home visits and spend more time in sharing experiences. They also expressed their interest to meet other breastfeeding mothers living in the same neighborhood, something that was not possible through the trial. Moreover, they preferred to have met their peer supporters during the trial's prenatal session that they attended.

The peer supporters used the knowledge and skills acquired during their participation in the trial to reach out to pregnant, and postpartum women outside the trial. They were some-times put in contact with pregnant, or breastfeeding women through their family members and friends, with the aim of providing information on breastfeeding. These are examples of personal efforts that peer supporters engaged in, beyond their role in the trial:

*My friends started calling me up asking about tips for breastfeeding because they know I am supporting other women.*

*(33 years old, two children)*

*I feel like I am serving the society with the things I know... when you can make a difference with your knowledge it is encouraging to keep doing the same whenever you are needed.*

*(29 years old, two children)*

## Becoming breastfeeding advocates

Similar to peer supporters, breastfeeding mothers reported being engaged in discussions with other women in their social circles, where they disseminated information about breastfeeding, corrected misconceptions, and gave tips:

*I told my friends about breastfeeding; some agreed with me, others disagreed especially because of having to wake up at night to feed the infant. I started giving advice to women about how to do it (breastfeeding), everything I learned from the study. They call me sometimes to ask me. I became a peer supporter without being one in the study. This is not from my experience but from what I learned from the peer supporter and the lactation consultant.*

*(24 years old, one child)*

*I started to behave as a peer supporter for three of my friends who gave birth. I encouraged them not to stop, and even when one of them wanted to stop, I insisted on helping her until she resumed breastfeeding. I had the moral support, as well as the information, so that I helped others as I had received help myself.*

*(27 years old, one child)*

Participants often found themselves in situations where they had to defend their breastfeeding choices, and provided information about the scientific evidence to other women who doubted their choice:

*There are things that the social circle tells you: give formula to the infant, give pacifier (dummy), you don't have enough milk, the infant will not grow with mother's milk only. I answer with what I learned from the study and tell them this is science. I have stronger arguments now.*

*(24 years old, one child)*

Peer supporters reported benefiting from their experience in the trial. They valued the training they received from the trial team, and especially the acquired knowledge, which corrected a number of common misconceptions about breastfeeding.

*I am a trained nurse and I thought that I knew everything about breastfeeding, but I discovered that there is a lot of information that I had missed throughout my training, or even social norms that I took for granted. The training given through this study was very beneficial for me....I didn't know that you can keep breast milk for up to 6 months in the freezer!*

*(30 years old, two children)*

Peer supporters explained how gaining information and communication skills enticed them to apply these skills beyond the study participants. The newly acquired knowledge and skills complemented their initial interest in breastfeeding. They named themselves as "breast-feeding advocates" who were gaining more recognition in their families, among their friends, and in their neighborhoods. One peer supporter for example, shared her ambition to learn more about breastfeeding, and improve her skills in order to engage fully in breastfeeding advocacy efforts. Peer supporters perceived that their involvement in the trial gave them a first-hand experience with the need to support breastfeeding mothers, and reinforced their commitment towards advocating exclusive breastfeeding:

*I was always pro-breastfeeding and I gained the scientific background and skills through this study.*

*(28 years old, two children)*

*Now, wherever I am, I look for an opportunity to talk about breastfeeding. I feel like I need to do this.. to do this good thing for everyone's sake. I talk to pregnant women, to women I see bottle feeding, everyone. I think I can help them and I want to do this even after the study stops.*

*(34 years old, three children)*

## Discussion

This study explored the process of support as viewed by breastfeeding mothers and peer supporters from a middle income country that lacks breastfeeding promotion and support structures within its health care system. Given the evidence supporting peer, and professional breastfeeding support interventions [1], our findings provide some explanations about the process through which different types of support influence breastfeeding outcomes, and shed light on the intervention processes which will contribute to the interpretation of trial outcomes [4]. In particular, it provides insight into how peer and IBCLC support contributed to the improvement of the six-month exclusive breastfeeding rate in our trial [14].

Our participants valued peer and IBCLC support, especially in boosting their perceived ability to cope with a demanding situation. They used the emotional support provided by peer supporters to deal with emotional challenges, and the instructional support given by IBCLC to overcome medical problems that breastfeeding mothers face. However, they seemed to have a preference for the type of support provided by the IBCLCs, mostly because it was provided face-to-face, and because it helped them overcome technical problems.

Participants noted the absence of face-to-face support from the peer supporters expecting that it would have enhanced their networks. Despite this disadvantage, participants received informational and emotional support from their peer supporters through verbal messages encouraging them to continue their efforts in breastfeeding. Positive relationships between breastfeeding mothers and peer supporters have been described as enhancing breastfeeding continuation through building trust, whereby accessing the peer at pivotal times during the postpartum period provides motivation to continue breastfeeding [22]. A recent study from Lebanon [23] highlighted the importance of peer support in facilitating breastfeeding practice of mothers in this context. Despite participants' perceptions of the benefits gained through peer support, this form of support did not enhance breastfeeding participants' social networks, mainly because the contact was limited to phone conversations.

The support provided by the IBCLCs was very important to breastfeeding mothers. It provided an alternative to clinic visits when faced with medical problems. Moreover, it was a trusted source of information that they needed and used. The fact that the communication with the IBCLCs was both by telephone and face-to-face, enhanced the positive nature of their relationship. Breastfeeding mothers were satisfied with the process through which IBCLC support was provided based on their demands, rather than on a fixed schedule only. The home visits provided by IBCLCs are valued mechanisms that help better understand breastfeeding mothers' social contexts, and provide emotional, instructional, information, and self-support forms of breastfeeding support [5].

Our findings reveal that the involvement of participants in peer support empowered them with knowledge and skills that they used beyond their involvement in the trial. The peer supporters identified themselves as breastfeeding advocates in their communities, a pattern that was previously observed in other contexts [8, 10]. Breastfeeding mothers also perceived themselves as peer supporters within their social circles, and endorsed that role with the new set of skills and knowledge they have earned during their participation in the trial.

Our study is limited by the fact that peer support ended up being delivered via telephone calls instead of a mixture of face-to-face and telephone support. This was mainly due to the security situation prevailing at that time, and to safety concerns of the peer supporters. Hence, the establishment of social networks between the breastfeeding mothers, and peer supporters may have been compromised. This probably contributed to breastfeeding mothers undervaluing peer support, and its potential beneficial effects on breastfeeding continuation. Another limitation is that participants who withdrew from the clinical trial, and who may have had negative experiences with peer support could not be interviewed. We also did not interview the IBCLCs in this study as we were aiming at developing an in-depth understanding of the peer support process. Since breastfeeding women's interviews highlighted the importance of IBCLC support, and drew comparisons between IBCLC and peer support, interviewing IBCLCs could have provided further understanding of the support process, and how best to use peer and IBCLC support to bolster breastfeeding practice.

## Conclusion

This study contributes to our understanding of women's perspectives with the process through which peer, and professional support are effective in promoting breastfeeding, and provides guidance for tailoring and scaling up of interventions in the Lebanese, and other similar contexts. As a qualitative study that was nested in a clinical trial, it provides an understanding of how breastfeeding mothers used the different forms of breastfeeding support interventions provided to them beyond the outcomes of the trial. Future interventions can focus on the implementation of this model through its adaptation to different hospital settings in the country. The findings also suggest that peer support systems influence more than breastfeeding outcomes, as they provide a structure for building a community of breastfeeding advocates. Enabling individuals with skills for peer support in breastfeeding has the potential to create supportive social networks in communities. The development of mother-to-mother support groups can be a potential low-cost resource to improve breastfeeding rates in similar populations. The provision of IBCLC support is highly dependent on the resources available in the health care system, and might be disregarded and compromised in a highly private system such as the one in Lebanon, especially in the absence of regulatory policies in this regard.

## Supporting information

**S1 Appendix. Question guide for in-depth interviews.**
(DOCX)

## Acknowledgments

We are thankful to our participants for their valuable contribution to our study.

## Author Contributions

**Conceptualization:** Tamar Kabakian-Khasholian, Mona Nabulsi.

**Data curation:** Tamar Kabakian-Khasholian, Hana Nimer, Soumaya Ayash, Fatima Nasser, Mona Nabulsi.

**Formal analysis:** Tamar Kabakian-Khasholian, Hana Nimer, Mona Nabulsi.

**Funding acquisition:** Mona Nabulsi.

**Investigation:** Tamar Kabakian-Khasholian, Hana Nimer, Mona Nabulsi.

**Methodology:** Tamar Kabakian-Khasholian, Mona Nabulsi.

**Project administration:** Mona Nabulsi.

**Supervision:** Tamar Kabakian-Khasholian.

**Validation:** Tamar Kabakian-Khasholian, Mona Nabulsi.

**Writing – original draft:** Tamar Kabakian-Khasholian, Hana Nimer, Mona Nabulsi.

**Writing – review & editing:** Tamar Kabakian-Khasholian, Hana Nimer, Soumaya Ayash, Fatima Nasser, Mona Nabulsi.

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
