## [Decision Letter · Decision Letter 0]

15 Jul 2019

PONE-D-19-15647

Experiences with peer and professional support for breastfeeding in Beirut, Lebanon: A qualitative study

PLOS ONE

Dear Dr Nabulsi,

Thank you for submitting your manuscript to PLOS ONE. After careful consideration, we feel that it has merit but does not fully meet PLOS ONE’s publication criteria as it currently stands. Therefore, we invite you to submit a revised version of the manuscript that addresses the points raised during the review process.

We would appreciate receiving your revised manuscript by Aug 29 2019 11:59PM. To enhance the reproducibility of your results, we recommend that if applicable you deposit your laboratory protocols in protocols.io, where a protocol can be assigned its own identifier (DOI) such that it can be cited independently in the future. For instructions see: http://journals.plos.org/plosone/s/submission-guidelines#loc-laboratory-protocols

We look forward to receiving your revised manuscript.

Kind regards,

Jennifer Yourkavitch

Academic Editor

PLOS ONE

Journal Requirements:

Reviewers' comments:

Reviewer's Responses to Questions

**Comments to the Author**

1. Is the manuscript technically sound, and do the data support the conclusions?

Reviewer #1: Yes

Reviewer #2: No

2. Has the statistical analysis been performed appropriately and rigorously? 

Reviewer #1: N/A

Reviewer #2: I Don't Know

3. Have the authors made all data underlying the findings in their manuscript fully available?

Reviewer #1: Yes

Reviewer #2: Yes

4. Is the manuscript presented in an intelligible fashion and written in standard English?

Reviewer #1: Yes

Reviewer #2: Yes

5. Review Comments to the Author

Reviewer #1: Thank you for the opportunity to review this manuscript about the experiences of mothers and peer supporters who participated in a breastfeeding promotion trial in Lebanon. Understanding the experiences of peer supporters and the women they support is an important contribution to the literature particularly in middle-income country settings.

In general, the manuscript would be strengthened by including more information about the intervention and some reference to the trial outcomes. If the intervention improved breastfeeding practices, knowing that would help the reader to put these findings in context. While there are references to the trial protocol paper, having more details about the intervention in this manuscript would be helpful to readers.

Did you conduct interviews with the IBCLCs?

My specific comments are below:

Line 13 – in the abstract and the introduction, there is reference to a range of initiation rates, what accounts for this range?

Line 22 – if software was used for analysis please add here, also, “informed by the principles of grounded theory” could also be added to this sentence.

Line 23 – Is “their experience” referring to experience receiving professional and peer lactation support, or breastfeeding in general or participating in the trial? Please clarify

Line 24 – “lactation support provider” is confusing especially compared to peer support, clarifying that this was professional lactation support provided by IBCLCs should be clearly stated. There are other places in the manuscript where this is not clear, please make this change throughout so that anywhere that refers to professional support it is clear that it is the trial IBCLCs.

Line 51 – Please note that Sousa et al identified 5 different types of support for breastfeeding women

Line 53-66 – It would be helpful to clearly state that social support and network influences can also negatively influence BF and other health practices if not involved/supportive of recommended practices

Line 67 – consider a different word for “Bonding”

Line 68-69 – please note that that bridging women from different backgrounds was specific to the study in England. Peer educators typically reflect the backgrounds of women in their communities

Line 72 – please clarify if social networking refers to online/social media

Line 75 – providing a platform for employment really varies in the lay health worker/volunteer literature – consider adding “can” or “may” serve as a platform

Line 90 – please include something about the results of the trial? Is this paper out? It would be very helpful to know if the intervention had an impact or not? If it did not, exploring more about that would also be helpful.

Line 105-112 – having a figure or table that describes all of the elements of the intervention would be very helpful. Who delivered the prenatal education, and where and when was it delivered? Make it clear here that postnatal phone calls by peer supporters were provided and there were no in-person visits (if that is correct). Were peer supporters compensated in any way? How much time did they spend providing support? How many women did each IBCLC support? How many IBCLCs were there?

Line 123 – How many women did each peer supporter support? if women identified someone to be their peer supporter, did that supporter provide support to other women or only to her? Were peer supporters assigned to multiple women, some of whom they knew and others they did not?

Line 132 – is there a reference for the LOVE approach?

Line 142 – the timing of breastfeeding support is unclear – above it sounded like only postnatal peer support was provided but here was it also prenatal? Having some sort of figure that showed IBCLC and peer contacts and timing would be helpful. Could mothers call their peer whenever they wanted?

Line 156 – please include peer supporters support a range of x-x women with most supporting x or something like that

Line 157-166 – Some of this could go in the setting description

Line 166 – if IBCLCs are not available at either hospital, do IBCLCs have private practices or how is that structured in Beirut?

Line 167 – Are there policies in Lebanon related to breastfeeding? Has the code of marketing of breastmilk substitutes been implemented?

Line 168 – what does not organized mean?

Line 214 – Was qualitative analysis software used?

Results section - For all quotes, please include attributions, such as x year old mother with # children, peer supporter with attributes you think are important and can distinguish respondents from each other

Line 287 – this is not clear, would they have wanted to have prenatal contacts or did some have prenatal peer support and others not?

Line 353 – also IBCLCs providing home visits, right?

Line 360 – did mothers prefer support from IBCLC? Is sounds like it from the results?

Line 366 – and emotional support?

Line 384- since women received support from IBCLCs and peer supporters, and based on the results presented, it is hard to distinguish the impact of the support from peers from IBCLCs – so this perhaps should refer to support in general. The fact that women received both professional and peer support could be discussed more and additional literature could be referenced.

Line 398 – Please include your thoughts about the implications of this research for the design of future interventions to support breastfeeding or for additional research that is needed. For example, are there alternative ways to reach women - using social media platforms for example – that might be appropriate to test? Or when security is less of an issue would support groups or other delivery platforms work? Since it sounds like women wanted to connect with other mothers something like La Leche League model or other mother-to-mother support groups may be helpful. Also, based on your findings would you recommend both types of support to women? If a health system has limited resources should they invest in both IBCLC and peer support?

Line 400 – peer and professional support?

Reviewer #2: This is an interesting report, addressing the important issue of breastfeeding support.

My comments are:

Abstract:

Pg 2, Line 13. The exclusive breastfeeding (EBF) rate mentioned here is "15% by 6 months". Unicef SOW writes it as "15% for <6 months". This has a different meaning. If you have any data for EBF at 6 months, it would be useful.

Line 16. AIM: "To describe the experiences of breastfeeding mothers and peer support providers with the process of peer support...." It does not include the "professional support" or the "lactation support providers" that are mentioned in your Results (line 24).

Methods. This section is not clear. There is no mention of "professional/ expert/ trained lactation support providers", although the Title of the paper is "Experiences with peer and professional support for breastfeeding.......". Were the "lactation support providers" not interviewed? If yes, they should be included here. And if not, then the Title would need to change, and other sections modified accordingly.

There is no mention of this group in lines 90-91 of the Introduction either. This makes the whole paper, including the Design and the Results confusing. The design of the main trial was different. This particular cross-sectional qualitative study should focus totally on the interviews conducted at 6 months - with breastfeeding mothers, peer supporters, and the lactation support providers/ IBCLCs mentioned later.

I would the authors to also include a snapshot of the Setting in this paper as well - not only to refer to Reference #10.

The total no. of peer supporters was 39 - but only 21 were interviewed? How many IBCLCs were included in the trial? Were they full time/ part employees of the hospitals? Or were they in private practice? Were they paid by the project? If yes, how much? Per month/ per visit? These questions are important from the sustainability point of the project. If this group is included in this paper, then Table #1 should then also include the characteristics of these professionals.

The quotes are good. Discussion and Conclusions would need to change. Peer support (Line 400) was only telephonic peer support, whereas professional support by IBCLCS (not mentioned here) was face-to-face through home visits. I think these two forms of support are totally different and actually should not be compared! Peer support in many countries is also face-to-face.

6. PLOS authors have the option to publish the peer review history of their article (what does this mean?). If published, this will include your full peer review and any attached files.

Reviewer #1: No

Reviewer #2: No

---

## [Author Response · Author response to Decision Letter 0]

26 Aug 2019

Journal Requirements

1. Ensure that your manuscript meets PLOS ONE’s style requirements, including those for file naming.

Answer: Done.

2. Include captions for your Supporting Information files at the end of your manuscript, and update any in-text citations to match accordingly.

Answer: Done. 

Response to Reviewer #1

We greatly appreciate your critical valuable comments and suggestions. We hope that our response addresses each of the raised points/comments. 

1. In general, the manuscript would be strengthened by including more information about the intervention, and some reference to the trial outcomes. If the intervention improved breastfeeding practices, knowing that would help the reader to put these findings in context. While there are references to the trial protocol, having more details about the intervention in this manuscript would be helpful to readers.

Did you conduct interviews with the IBCLCs?

Answer: In the Introduction (Page 6, lines 87-95), we added the following general description of the intervention: “A multidisciplinary group of researchers at the American University of Beirut launched a multi-component breastfeeding support intervention consisting of breastfeeding education, peer support, and professional lactation support delivered by International Board Certified Lactation Consultants (IBCLCs) to target this gap”. We also added on page 7, lines 93-95 the following statement about the primary outcome result: “The multi-component intervention was effective in improving the rate of six-month exclusive breastfeeding in the intervention group by two folds, as compared to standard care [14]”. 

In this qualitative study, we did not conduct interviews with IBCLCs since the main aim of the study was to describe the experiences of breastfeeding mothers and their peer supporters with the process of breastfeeding support. Breastfeeding mothers however drew comparisons between the different types of support they received which we had to include in our results. We clarified in the Methods/Sample section that IBCLCs were not interveiwed.

2. Line 13: In the abstract and introduction, there is reference to a range of initiation rates, what accounts for this range?

Answer: We had reported a range in breastfeeding rates based on different sources of data. In the absence of large population-based studies in Lebanon, these are all estimates from small scale research projects. In the revised manuscript, we kept the rate that was published by UNICEF in 2015 (15% in infats below six months) to avoid confusion. This was done both in the Abstract (lines 12-13), and in the Introduction (lines 84-85). 

3. Line 22: If software was used for analysis please add here, also, “informed by the principles of grounded theory” could also be added to this sentence.

Answer: Analysis was done manually without the use of software. We added the suggested phrase on line 23-24 in the Abstract. 

4. Line 23: Is “their experience” referring to experience receiving professional and peer lactation support, or breastfeeding in general, or participating in the trial? Please clarify. 

Answer: The term “experience” refers to the experience with breastfeeding. We clarified this point on line 25. 

5. Line 24: “Lactation support provider” is confusing especially compared to peer support. Clarifying that this was professional lactation support provided by IBCLCs should be clearly stated. There are other places in the manuscript where this is not clear. Please make this change throughout so that anywhere that refers to professional support it is clear that it is the trial IBCLCs.

Answer: We made the necessary changes throughout the manuscript to clarify that IBCLCs refer to professional lactation support providers. 

6. Line 51: Please note that Sousa et al identified 5 different types of support for breastfeeding women. 

Answer: The original publication by Sousa et al is available to us in Portuguese only. Because of the language barrier, we used a secondary source available in English (Oliveira et al, 2017) which is a review article where these concepts are used and discussed. We added Sousa et al as an additional citation as we had access to the abstract in English. 

7. Lines 53-66: It would be helpful to clearly state that social support and network influences can also negatively influence BF and other health practices if not involved/supportive of recommended practices.

Answer: We made adjustments to the sentence on lines 62-64 to capture the meaning suggested by the reviewer: “The relationships in a network that support recommended practices enhance health and well-being by overcoming the stressors, and enhancing the individual’s ability to access new contacts and information; and to identify and manage problems”.

8. Line 67: Consider a different word for “Bonding”. 

Answer: We change the term “bonding” to the term “connecting” (line 70). 

9. Lines 68-69: Please note that bridging women from different backgrounds was specific to the study in England. Peer educators typically reflect the backgrounds of women in their communities. 

Answer: We clarified this point through some editing on line 71. 

10. Line 72: Please clarify if social networking refers to online/social media.

Answer: This refers to online social networking (line 76). 

11. Line 75: Providing a platform for employment really varies in the lay health worker/volunteer literature – consider adding “can” or “may” serve as a platform. 

Answer: We agree with the reviewer. We added “may” to the sentence on line 78. 

12. Line 90: Please include something about the results of the trial. Is this paper out? It would be very helpful to know if the intervention had an impact or not. If it did not, exploring more about that would also be helpful.

Answer: When we submitted this manuscript, the paper presenting the findings from the trial was not yet published. The paper has been published recently in PLOS ONE, and we have referred to it in the revised mansucript (Lines 93-95).

13. Lines 105-112: Having a figure or table that desribes all the elements of the intervention would be very helpful. Who delivered the prenatal education, and where and when was it delivered? Make it clear here that postnatal phone calls by peer supporters were provided and there were no in-person visits (if that is correct). Were peer supporters compensated in any way? How much time did they spend providing support? How many women did each IBCLC support? How many IBCLCs were there?

Answer: We have published a figure in the paper that describes our trial findings detailing the multi-component intervention (Nabulsi et al. PLoS ONE. 2019; 14(6): e0218467). In the revised manuscript, we added the following statements to address the above questions of the kind reviewer: “Participants in the experimental group received a multi-component intervention consisting of: (a) prenatal breastfeeding education delivered by IBCLCs in one of the two trial centers as prenatal classes to raise awareness, and improve knowledge, (b) postpartum lactation support provided by IBCLCs to improve skills, and self-efficacy through home visits, (c) postpartum peer (lay) telephone support to build social support, and enhance social capital. The primary outcome of the trial, exclusive breastfeeding, was assessed at the six month interview with the mothers. More details on the trial’s multi-component intervention and the trial’s results can be found elsewhere [13, 14]”. 

We are referring to the trial publication instead of providing a figure about the intervention to avoid ‘self-plagiarism’. 

As for the remaining questions, peer supporters were compensated for their telephone calls. They were asked to provide at least 10 phone calls starting day of delivery and up to six months. The time spent providing peer support was very variable among the different participants (range: 0-30 minutes/call). We had 7 IBCLCs that provided professional lactation support to our participants. The number of women per each IBCLC was also variable, depending on availability of IBCLC and ranged between 12 and 52 breastfeeding mothers/IBCLC. 

14. Line 123: How many women did each peer supporter support? If women identified someone to be their peer supporter, did that supporter provide support to other women or only to her? Were peer supporters assigned to multiple women, some of whom they knew and others did not?

Answer: We added the following sentence to address this comment (lines 131-134): “In total, 39 peer supporters were recruited into the trial. The number of breastfeeding mothers assigned to the same peer supporter ranged between 1 and 9, depending on availability of peer supporter. Some of the breastfeeding mothers were known to peer supporters through their social networks, while others were not”. 

15. Line 132: Is there a reference for the LOVE approach?

Answer: We added the reference for the LOVE approach to the revised manuscript (reference 17).

16. Line 142: The timing of breastfeeding support is unclear – above, it sounded only postnatal peer support was provided, but here was it also prenatal? Having some sort of figure that showed IBCLC and peer contacts and timing would be helpful. Could mothers call their their peer whenever they wanted?

Answer: Breastfeeding mothers were assigned to their peer supporters during the prenatal breastfeeding education class. The peer supporter would then call the breastfeeding mother to introduce herself. The actual peer support started on the first day postpartum. Peer supporters were given the option of communicating by phone or through home visits. They all opted for the phone call approach throughout the study. The recommended schedule of contact for the peer supporters in the trial is described on lines 152-160, and that of the IBCLCs is added to lines 161-168.

17. Line 156: Please include peer supporters support a range of X-x women with most supporting x, or something like that.

Answer: This information is now included in lines 132-135.

18. Lines 157-166: Some of this could go in the setting description.

Answer: We agree with the reviewer and we moved some of the text to the Setting section (lines 181-188). 

19. Line 166: If IBCLCs are not available at either hospital, do IBCLCs have private practices or how is that structure in Beirut?

Answer: IBCLCs have private practices in Lebanon. There are few IBCLCs in the country and they are not employed by the hospitals. This information is added on lines 187-188.

20. Line 167: Are there policies in Lebanon related to breastfeeding? Has the code of marketing of breast milk substitutes been implemented?

Answer: We added a description of breastfeeding related policies in Lebanon to the Setting section (lines 189-192). 

21. Line 168: What does not organized mean?

Answer: We meant that they did not follow standards of practice. We amended the sentence for clarity (line 175). 

22. Line 214: Was qualitative analysis software used?

Answer: We conducted the analysis manually without the use of software (line 229). 

23. Results section: For all quotes, please include attributions, such as x year old mother with # children, peer supporter with attributes you think are important and can distinguish respondents from each other.

Answer: These are now added on all quotes used in the Results section. 

24. Line 287: This is not clear. Would they have wanted to have prenatal contacts, or did some have prenatal peer support and others not?

Answer: As mentioned in our answer to comment 16 above, breastfeeding mothers were assigned to their peer supporters during the prenatal breastfeeding education class. Subsequently, the peer supporter would call the breastfeeding mother to introduce herself. The actual peer support started on the first day postpartum. Peer supporters were given the option of communicating by phone or through home visits. They all opted for the phone call approach throughout the study. The reason for this was that shortly after the trial launched, the security situation in Beirut deteriorated rapidly with several terrorist acts happening such as explosions and bombs. It was decided to leave it up to the peer supporters to decide on whether it would be safe to conduct face to face visits with breastfeeding mothers or to provide support via telephone calls. 

25. Line 353: Also IBCLCs providing home visits, right?

Answer: Yes, IBCLCs provided home visits. 

26. Line 360: Did mothers prefer support from IBCLCs? It sounds like it from the results.

Answer: We agree with the reviewer that overall there was a preference for the professional support provided by IBCLCs, although women also appreciated the peer support and found it useful for their overall experience. We mentioned their preference on lines 403-405. 

27. Line 366: and emotional support?

Answer: Yes, also emotional support. This is now added on line 394. 

28. Line 384: Since women received support from IBCLCS and peer supporters, and based on the results presented, it is hard to distinguish the impact of the support from peers and from IBCLS – so this perhaps should refer to support in general. The fact that women received both professional and peer support could be discussed more, and additional literature could be referenced.

Answer: We agree with the reviewer that breastfeeding women benefited from both peer and IBCLC support. However,this section (lines 412-417) discusses the findings that are relevant to peer supporters, and not the support received by women in general. Moreover, breastfeeding mothers who gained breastfeeding knowledge and skills from their IBCLCs wanted to replicate the role of their peer supporters by supporting other breastfeeding mothers in their social circle. 

29. Line 398: Please include your thoughts about the implications of this research for the design of future interventions to support breastfeeding, or for additional research that is needed. For example, are there alternative ways to reach women – using social media platforms for example that might be appropriate to test. Or, when security is less of an issue, would support groups or other delivery platforms work? Since it sounds like women wanted to connect with other mothers, something like La Leche League model or other mother-to-mother support groups may be helpful. Also, based on your findings, would you recommend both types of support to women? If a health system has limited resources, should they invest in both IBCLC and peer support?

Answer: Implications for research and practice are added to the conclusion section (lines 443-447). 

30. Line 400: peer and professional support?

Answer: Yes, we agree with the reviewer. The is revised to include professional support as well (line 434). 

Response to Reviewer #2

Thank you for your positive feedback and valuable comments. Please find below our point by point response. 

1. Abstract, Page 2, Line 13: The exclusive breastfeeding (EBF) rate mentioned here is “15% by 6 months”. UNICEF SOW writes it as “15% for <6 months”. This has a different meaning. If you have any data for EBF at 6 months, it would be useful.

Answer: We amended the text to correspond to the used definition in the cited UNICEF report “exclusive breastfeeding is only 15% in infants below six months” in the Abstract (lines 12-13), and in the Introduction (lines 84-85). 

2. Abstract, Line 16: AIM: “To describe the experiences of breastfeeding mothers and peer support providers with the process of peer support..” It does not include the “professional support” or the “lactation support providers” that are mentioned in your Results (line 24). 

Answer: The aim of the study is to describe the experiences of breastfeeding mothers and peer supporters with the process of breastfeeding support. We did not interview IBCLCs but breastfeeding women described their experience with both types of support sources they had in the trial. We made a slight adjustment to the aim stated in the abstract to clarify this point (line 17). 

3. Methods: This section is not clear. There is no mention of “professional/expert/trained lactation support providers”, although the Title of the paper is Experiences with peer and professional support for breastfeeding...”. Were the “lactation support providers” not interviewed? If yes, they should be included here. And if not, then the Title would need to change, and other sections modified accordingly.

Answer: IBCLCs were hired by the trial to provide professional breastfeeding support. They were not interviewed in this qualitative study because our aim was to describe the process of providing and receiving peer support. We clarified this point on lines 207-208. However, during the interviews, breastfeeding mothers drew comparisons between peer and IBCLC support as both were provided in the trial. Thus the results section highlighted these comparisons as an emerging theme from the data. We removed the word ‘professional’ from the title as suggested by the reviewer. 

There is no mention of this group in lines 90-91 of the Introduction either. This makes the whole paper, including the Design and the Results confusing. The design of the main trial was different. This particular cross-sectional qualitative study should focus totally on the interviews conducted at 6 months – with breastfeeding mothers, peer supporters, and the lactation support providers/IBCLCs mentioned later.

Answer: This is the same comment addressed above. 

I would advise the authors to also include a snapshot of the Setting in this paper as well – not only to refer to Reference #10.

Answer: We have included a description of the trial in this paper in as much space is allowed. We also referred to 2 previous publications about the trial (the protocol and the findings). Kindly refer to the information on the Setting, pages 12-13.

The total no. of peer supporters was 39- but only 21 were interviewed? How many IBCLCs were included in the trial? Were they full time/ part time employees of the hospital? Or were they in private practice? Were they paid by the project? If yes, how much? Per month/ per visit? These questions are important from the sustainability point of the project. If this group is included in this paper, then Table 1 should then also include the characterisitcs of these professionals.

Answer: We selected a purposive sample of peer supporters and used the thematic saturation principle to achieve the required sample size. Thus we were not in need of interviewing all peer supporters once we reached saturation with 21 peer supporters. As for the IBCLCs, we had 7 IBCLCs that provided professional lactation support to our participants. They were hired to do the support and were paid per visit ($50/hour). In Lebanon, IBCLCs are in private practice and are not yet integrated within the health care system. As mentioned above, IBCLCs were not interviewed in this study since our main aim was about experiences with peer support. 

The quotes are good. Discussion and Conclusions would need to change. Peer support (Line 400) was only telephone support, whereas professional support by IBCLCs (not mentioned here) was face-to-face through home visits. I think these two forms of support are totally different and actually should not be compared! Peer support in many countries is also face-to-face. 

Answer: We made some changes to the conclusion section adding implications for research and practice. We agree with the kind reviewer that the 2 forms of support are different. In this paper, we are not trying to compare and contrast between the 2 forms. However, this comparison between peer and professional support was made by the breastfeeding mothers and we had to include it in our findings since it emerged from the data. 

4. Table 1: Use footnote to clarify what the asterix and pound symbols mean.

Answer: There are no asterix or pound symbols in Table 1. 

5. Throughout: address punctuation to ensure clarity in sentences. E.g., line 37 (insert comma before and after ‘at six months’).

Answer: Done.

---

## [Decision Letter · Decision Letter 1]

26 Sep 2019

Experiences with peer support for breastfeeding in Beirut, Lebanon: A qualitative study

PONE-D-19-15647R1

Dear Dr. Nabulsi,

We are pleased to inform you that your manuscript has been judged scientifically suitable for publication and will be formally accepted for publication once it complies with all outstanding technical requirements.

With kind regards,

Jennifer Yourkavitch

Academic Editor

PLOS ONE

Additional Editor Comments (optional):

Reviewers' comments:

Reviewer's Responses to Questions

**Comments to the Author**

1. If the authors have adequately addressed your comments raised in a previous round of review and you feel that this manuscript is now acceptable for publication, you may indicate that here to bypass the “Comments to the Author” section, enter your conflict of interest statement in the “Confidential to Editor” section, and submit your "Accept" recommendation.

Reviewer #1: All comments have been addressed

2. Is the manuscript technically sound, and do the data support the conclusions?

Reviewer #1: Yes

3. Has the statistical analysis been performed appropriately and rigorously? 

Reviewer #1: N/A

4. Have the authors made all data underlying the findings in their manuscript fully available?

Reviewer #1: (No Response)

5. Is the manuscript presented in an intelligible fashion and written in standard English?

Reviewer #1: Yes

6. Review Comments to the Author

Reviewer #1: The authors comprehensively addressed my comments and revised their manuscript accordingly. I do not have any other comments.

7. PLOS authors have the option to publish the peer review history of their article (what does this mean?). If published, this will include your full peer review and any attached files.

Reviewer #1: No

---

## [Editor Report · Acceptance letter]

1 Oct 2019

PONE-D-19-15647R1 

Experiences with peer support for breastfeeding in Beirut, Lebanon: A qualitative study 

Dear Dr. Nabulsi:

I am pleased to inform you that your manuscript has been deemed suitable for publication in PLOS ONE. Congratulations! Your manuscript is now with our production department. 

With kind regards,

on behalf of

Dr. Jennifer Yourkavitch 

Academic Editor

PLOS ONE